# Endothelial Cell Response in Kawasaki Disease and Multisystem Inflammatory Syndrome in Children

**DOI:** 10.3390/ijms241512318

**Published:** 2023-08-01

**Authors:** Jihoon Kim, Chisato Shimizu, Ming He, Hao Wang, Hal M. Hoffman, Adriana H. Tremoulet, John Y.-J. Shyy, Jane C. Burns

**Affiliations:** 1Department of Biomedical Informatics, University of California, San Diego, CA 92093, USA; 2Section of Biomedical Informatics and Data Science, Yale School of Medicine, New Haven, CT 06510, USA; 3Department of Pediatrics, University of California, San Diego, CA 92093, USA; 4Department of Medicine, University of California, San Diego, CA 92093, USA; 5Rady Children’s Hospital, San Diego, CA 92123, USA

**Keywords:** Kawasaki disease, MIS-C, endothelial cell, WGCNA, network analysis, NFκB pathway, apoptosis, autophagy, EndoMT, RNA-seq

## Abstract

Although Kawasaki disease (KD) and multisystem inflammatory syndrome in children (MIS-C) share some clinical manifestations, their cardiovascular outcomes are different, and this may be reflected at the level of the endothelial cell (EC). We performed RNA-seq on cultured ECs incubated with pre-treatment sera from KD (*n* = 5), MIS-C (*n* = 7), and healthy controls (*n* = 3). We conducted a weighted gene co-expression network analysis (WGCNA) using 935 transcripts differentially expressed between MIS-C and KD using relaxed filtering (unadjusted *p* < 0.05, >1.1-fold difference). We found seven gene modules in MIS-C, annotated as an increased TNFα/NFκB pathway, decreased EC homeostasis, anti-inflammation and immune response, translation, and glucocorticoid responsive genes and endothelial–mesenchymal transition (EndoMT). To further understand the difference in the EC response between MIS-C and KD, stringent filtering was applied to identify 41 differentially expressed genes (DEGs) between MIS-C and KD (adjusted *p* < 0.05, >2-fold-difference). Again, in MIS-C, NFκB pathway genes, including nine pro-survival genes, were upregulated. The expression levels were higher in the genes influencing autophagy (*UBD*, *EBI3*, and *SQSTM1*). Other DEGs also supported the finding by WGCNA. Compared to KD, ECs in MIS-C had increased pro-survival transcripts but reduced transcripts related to EndoMT and EC homeostasis. These differences in the EC response may influence the different cardiovascular outcomes in these two diseases.

## 1. Introduction

Kawasaki disease (KD) and multisystem inflammatory syndrome in children (MIS-C) share some similar signs and symptoms [1]. Although patients with both diseases are highly inflamed, a disease classification algorithm using age, five clinical signs and 17 clinical laboratory values was able to distinguish MIS-C from KD and other febrile illnesses with a >90% accuracy [2]. Increased inflammatory markers were among the key features in the classification algorithm that differentiated MIS-C from KD. The standard treatment for KD patients is intravenous immune globulin (IVIG), and 80–85% of patients respond to this treatment with cessation of fever [3,4]. In contrast, MIS-C patients are also treated with IVIG but frequently require additional anti-inflammatory therapy, including steroids and the blockade of tumor necrosis factor (TNF)-α and interleukin (IL)-1 [5].

The cardiovascular involvement in KD and MIS-C differs with transmural coronary artery inflammation, resulting in coronary artery aneurysms associated only with KD. MIS-C, in contrast, commonly presents with decreased left ventricular contractility that reverses with anti-inflammatory therapy with no long-term sequelae in most patients [6]. There are only a limited number of autopsy reports describing clear cases of MIS-C; however, there is no long-term clinical vascular pathology associated with MIS-C [7,8]. Molecular studies have compared and contrasted KD and MIS-C at the transcriptional and proteomic levels in circulating blood [9]. The upregulation of inflammatory pathways in both KD and MIS-C has underscored the similarities between the two conditions. We sought to characterize KD and MIS-C at the level of the EC to determine if the EC response might better reflect the divergent clinical outcomes of the two diseases.

To interrogate the EC response in patients with KD, we previously developed an experimental system using cultured endothelial cells (ECs) incubated with sera from pre-treatment KD patients [10]. Using that system, we demonstrated that ECs incubated with KD sera expressed a transcriptional profile associated with endothelial–mesenchymal transition (EndoMT) and altered EC homeostasis with reduced levels of nitric oxide synthase 3 (NOS3) [10]. To unravel the molecular mechanism of the divergent cardiovascular outcomes in KD and MIS-C, we studied transcriptional profiles of cultured ECs incubated with acute, pre-treatment sera from patients with either KD or MIS-C using RNA-seq.

## 2. Results

### 2.1. RNA-seq and Differential Expression Analysis on Cultured ECs Incubated with Sera from the Patients with KD and MIS-C 

To understand the EC response in KD and MIS-C, sera from five KD patients, seven MIS-C patients, and three healthy controls (HC) were incubated individually with cultured human umbilical vein ECs. RNA-seq was performed on cultured EC lysates (Table 1, Figure 1). 

The first two principal components (PCs) of the RNA-seq data showed a clear separation of the three experimental groups, with the healthy control samples indistinguishable from the no-sera samples (Figure 2A).

### 2.2. Weighted Gene Co-Expression Analysis (WGCNA)

The gene co-expression network was constructed with 935 differentially expressed genes (DEGs) between KD and MIS-C, partitioned into seven gene modules. Each module was labeled with a unique color name, with the gray color module as the least connected genes following the WGCNA color scheme. The topological-overlap matrix (TOM) plot (Figure 2B) shows the interconnectivity pattern among seven modules. The rows and columns of the TOM plot are genes, the modules correspond to blocks of highly connected genes, and the darker red color cells in the TOM plot represent higher interconnectedness. The three largest modules were turquoise > blue > brown (Figure 2C). To annotate the biological meaning for each module, all genes in each module were used to identify the enriched pathways (https://maayanlab.cloud/Enrichr/ (accessed on 28 July 2023) (Appendix A) and the top 10 hub genes with the highest intramodular connectivity (kWithin) (Appendix A) were selected. Based on this knowledge and data-driven findings, we annotated a biological function for each module (Figure 3A). 

The largest module (turquoise, Figure 3B) was annotated as the “TNFα/NFκB” pathway, and its top 10 hub genes included *RELB proto-oncogene, Nuclear factor kappa B (NFκB) subunit* (*RELB*), *sequestosome 1* (*SQSTM1*), and *Nuclear factor kappa B subunit 1* (*NFKB1*) (Appendix A). Many turquoise module genes belonged to the TNFα and IL1 pathways from the knowledge-based analysis and corresponded to the top hub genes from the data-driven network since NFκB is the main downstream pathway regulated by TNFα and IL1 (Figure 2D). 

The second largest module (blue) was annotated as EC homeostasis. The hub genes of the blue module included *endomucin* (*EMCN*), *RUNX1 partner transcriptional co-repressor 1* (*RUNX1T1*), *nuclear factor I A* (*NFIA*), and *nuclear factor I B* (*NFIB*). The key genes in the enriched pathways for the blue module were the Brain-derived Neurotrophic Factor (BDNF) signaling pathway and NOS3-related pathways (Appendix A). NOS3 is the enzyme that synthesizes nitric oxide, the key molecule that maintains endothelial cell homeostasis. Its expression is tightly regulated by many transcription factors and molecules, including Nuclear Factor (NF)-1, the protein product of *NFIA* or *NFIB*, and glucocorticoids [11,12]. Vascular endothelial growth factor (VEGF) also synthesizes nitric oxide through eNOS activation [13], and EMCN, RUNX1T1, and BDNF modify the VEGF receptor pathway [14,15,16,17,18].

The third largest module (brown) had several ribosomal protein genes (*RPL27*, *RPL6*, and *RPS23*) as hub genes and was enriched with molecules in the translation pathway.

Despite the smallest number of genes, the red module had important hub genes and enriched pathways, including glucocorticoid (GC)-responsive genes and EndoMT suppression. Its hub genes included several GC-regulated genes, such as *hes family bHLH transcription factor 1* (*HES1*), *KLF transcription factor 10* (*KLF10*), *snail family transcriptional repressor 1* (*SNAI1*), *growth arrest, and DNA damage-inducible beta* (*GADD45B*) [19,20,21,22]. HES1 is a master regulator of glucocorticoid receptor-dependent gene expression. The apparent mutual antagonism of HES1 and the glucocorticoid (GC) receptor (GR) was reported [19]. SNAI is a key transcription factor to induce EndoMT through the BDNF-TrkB pathway [21]. 

The yellow module had no strongly enriched pathway with an adj *p* < 1 × 10^−3^. Its hub genes included anti-inflammatory genes, such as *peroxiredoxin-like 2A* (*PRXL2A*) [23] and *myocyte enhancer factor 2C* (*MEF2C*) [24] and genes related to the immune response, such as *lymphatic vessel endothelial hyaluronan receptor 1* (*LYVE1*) [25]. This module also includes the hub genes, *calcitonin receptor-like receptor* (*CALCRL*) [26], whose expression is regulated by GC and MEF2C [27], which cooperate with GC bound to GR to regulate gene expression. Therefore, this module may also have some connection with GC regulation. Similarly, the green and gray modules had no significant pathway with a *p* < 1 × 10^−3^. The hub genes in the green module included genes related to DNA repair, such as *X-ray repair cross-complementing 3* (*XRCC3*) [28] and *paired immunoglobin-like type 2 receptor beta* (*PILRB*) [29]. The hub genes in the green module included apoptosis-related genes, *zinc finger protein 428* (*ZNF428*) [30] and *modulator of apoptosis 1* (*MOAP1*) [31].

After annotating the biological meaning of each module, we further assessed module–module relations (Figure 3C) and module–trait relations (Figure 3B). For each module, boxplots by trait were created using the values of the module eigengene (Figure 3A), the first principal component of the expression matrix of the corresponding module. While the turquoise module genes were up in MIS-C (Figure 3B), the blue module genes were down in MIS-C. This opposite signal is reflected in the strong negative module–module correlation between turquoise and blue. The brown module genes were lower in MIS-C compared to KD.

The seven annotated modules were linked together visually to elucidate the contrasting underlying pathogeneses of the two diseases (Figure 3A).

### 2.3. Differential Expression Analysis with Rigorous Filter

Next, to directly identify the differences in the EC response between MIS-C and KD, a rigorous filter (MIS-C vs. KD fold difference > 2, adj *p* < 0.05) was applied. A comparison of transcript abundance in ECs incubated with sera from KD and MIS-C patients revealed only 41 DEGs (Appendix A). The most significant DEG between KD and MIS-C was CCL2 (2.6-fold increase in MIS-C, *p* = 7.4 × 10^−126^) (Figure 4A). Since CCL2 is secreted, the CCL2 protein levels were measured in the media in which the ECs were cultured. Although CCL2 transcript levels were very high in MIS-C compared to KD and HC, the secreted protein levels were not significantly different (Appendix A). 

Of the 41 DEGs, 31 (76%) belonged to the turquoise module (Figure 4B) and included TNF receptor-associated factor 1 (TRAF1), baculoviral IAP repeat containing 3 (BIRC3), TNFRSF4 and 9, TNF alpha-induced protein 3 (TNFAIP3), TNFAIP3 interacting protein 3 (TNIP3), and lymphotoxin beta (LTB). Other turquoise module genes included NFκB-regulated chemokine genes (CXCL1, 2, 3, 5, 6, CCL2, and 20) and the adhesion molecules intercellular adhesion molecule 1 (ICAM1), vascular cell adhesion molecule 1 (VCAM1), ADAM metallopeptidase domain 8 (ADAM8), and selectin E (SELE). All of these genes were upregulated in MIS-C compared to KD. Three molecules (Figure 4B, underlined genes) in the turquoise module were reported to bind to SQSTM1, the second top hub gene of the turquoise module (Figure 2D and Appendix A). These genes included Epstein–Barr virus-induced 3 (EBI3) and ubiquitin D (UBD), which are important in autophagy. Because the expression of genes in the NFκB pathway inhibits apoptosis by stimulating the expression of anti-apoptotic genes, we investigated whether the 41 DEGs were related to apoptosis. We found nine upregulated anti-apoptosis (pro-survival) genes and one downregulated pro-apoptosis gene (insulin-like growth factor-binding protein 5 (IGFBP5)) in MIS-C compared to KD (Figure 4B, blue and red arrows).

These 41 DEGs also included genes in the blue module (EC homeostasis), brown module (translation), red (glucocorticoids responsive gene and EndoMT suppression), and yellow (anti-inflammation and immune response) modules, all of which fit the biological functions annotated by WGCNA (Figure 5).

## 3. Discussion

The response of cultured ECs to incubation with acute, pre-treatment sera from KD and MIS-C patients differed significantly. ECs incubated with MIS-C sera expressed higher levels of transcripts associated with cell survival and lower levels of transcripts associated with EndoMT when compared to KD. ECs incubated with MIS-C sera had depressed levels of transcripts that are modulated by GC compared to KD, suggesting increased serum GC levels in MIS-C. These differences may influence the divergent cardiovascular outcomes in the two diseases (Figure 6). 

Activation of the NFκB pathway was critical in the EC response in MIS-C patients. NF-κB regulates the transcription of many pro-inflammatory and pro-survival genes. We found nine upregulated pro-survival genes in MIS-C compared to KD, including *S100A3* [51], *TNFRSF4* [52], *BIRC3* [53], *WWC1* [54], *VNN1* [55], *CD69* [56], *LTB* [57], *TRAF1* [58], and *ANO9* [59]. We also found that an independent pro-apoptotic gene, *IGFBP5* [60], was downregulated in MIS-C. Taken together, these findings suggest that ECs in MIS-C have a more pro-survival phenotype compared to KD. These findings are consistent with the report that circulating endothelial cells (CECs) were released into the blood in 100% of acute KD patients but in only 26% of acute MIS-C patients compared to the normal controls [50]. 

SQSTM1, also known as p62, is one of the key proteins induced by NFkB. SQSTM1 is a scaffold protein with multiple domains, which allows it to bind multiple proteins. By binding different molecules through these domains, SQSTM1 has pleiotropic effects, including activation of the NFκB pathway and mediating protein degradation (the ubiquitin–proteasome system and autophagy), cell death, and cell survival, depending on the binding proteins [61]. SQSTM1 and autophagy influence EndoMT, the process that transforms ECs into myofibroblasts [34,62]. Myofibroblasts clearly play a role in damaging the coronary arteries in KD [63,64]. 

Nitric oxide (NO) is a critical molecule in the maintenance of endothelial cell homeostasis and the regulation of vascular tone and permeability. Therefore, NO bioavailability is tightly regulated by NO synthases, especially NOS3 in ECs. TREH, CYP26B1, and GC also regulate NO syntheses. Overall, we found lower transcript levels of genes that regulate the NO levels in MIS-C. Abnormal EC homeostasis may be more severe in MIS-C compared to KD and may lead to myocardial stunning [49,65]. Takotsubo cardiomyopathy, a transient left ventricular dysfunction that spontaneously recovers within days, might be a model for myocardial dysfunction in MIS-C [49]. Of interest, genetic variants in BAG cochaperone 3 (BAG3), which has an important role in autophagy through binding to SQSTM1, have been associated with Takotsubo cardiomyopathy [66]. GC also induces vasospastic angina with elevated ST segments by affecting ECs [67]. GC decreases intracellular calcium mobilization, NOS3, and nitric oxide in ECs [12,68]. GC also suppresses the production of the vasodilator, prostacyclin, and increases the synthesis of the vasoconstrictor, thromboxane [69]. Abnormal EC homeostasis and GC-induced constriction could lead to reduced blood flow in the heart, contributing to myocardial dysfunction. LV dysfunction is a prominent feature of MIS-C [70,71,72,73]. The degree of endothelial dysfunction based on the studies of flow-mediated dilation of brachial arteries correlated with arterial stiffness and reduced ejection fractions in MIS-C [74].

GC-bound GR inhibits the expression of inflammatory molecules on the EC surface, such as adhesion molecules, SELE, and VCAM-1, and the secretion of chemokines, including CCL2 [44]. GCs control gene expression through transcriptional and post-transcriptional regulation. GC inhibits protein synthesis by inhibiting translation initiation and ribosomal synthesis at the level of transcription, post-transcription, and translation [46,47,48]. Translation initiation machinery includes many molecules including EIF3 and RPS6 kinase. These structures are composed of ribosomal RNA and ribosomal proteins (ribosome 40S containing 18S rRNA and 33 ribosomal proteins (RPS), and ribosome 60S containing three rRNAs (25S, 5.8S, 5S) and 49 ribosomal proteins (RPL)) [47,75,76]. The transcript levels of many of these ribosomal proteins were reduced in MIS-C compared to KD.

## 4. Limitations

In our descriptive study of the ex vivo EC response to sera from MIS-C and KD patients, we only characterized the transcriptome and not the translation of key transcripts into protein, except for CCL2, which was translated and secreted in the media. Limitations in the availability of pre-treatment patient sera precluded more extensive analyses. We justified the use of HUVECs instead of coronary artery ECs based on previous work by our group showing similar responses of these two cell lines [77]. However, it is possible that the responses could be different in this experimental model. A comparison of blood levels of GC between the patients is difficult since GC levels require standardized, timed blood samples that were not available before treatment.

## 5. Materials and Methods

### 5.1. Patients and Samples

The demographic and clinical characteristics of this study’s patients are presented in Table 1. All patients with KD and MIS-C were diagnosed by one of two clinicians specializing in KD and MIS-C (JCB and AHT) at Rady Children’s Hospital-San Diego and met the American Heart Association criteria for complete KD or CDC criteria for MIS-C [78,79]. In order to avoid the potential for misclassification, all MIS-C patients had positive antibody testing for the nucleocapsid protein of SARS-CoV-2, and none had received a SARS-CoV-2 vaccine. Serum samples were collected prior to treatment from the patients with KD (illness days 4–7) and MIS-C (illness days 2–8). Late convalescent (illness day 414–990 days) sera from KD with a remote history of KD and with always normal coronary arteries by echocardiography served as the healthy controls. 

### 5.2. Cell Culture 

The detailed methods were as previously described [10]. In the experiments involving patients’ sera, M199 was supplemented with 10% patient serum (individual) and 2% fetal bovine serum (FBS) (100 μL of patient serum, 20 μL of FBS and 880 μL of medium per well of a six-well plate)**.** For the RNA sequencing (RNA-seq), human umbilical vein ECs (HUVECs) were incubated with or without individual pre-treatment sera from KD patients, MIS-C patients, and HC for 24 h. 

### 5.3. RNA Extraction, RNA Sequencing, and RT-PCR

Total RNA from HUVECs was isolated using miRVana (ThermoFisher) for the RNA-seq. For RNA-seq analysis, RNA sequencing libraries were generated using the Illumina Ribo-Zero Plus rRNA Depletion Kit with IDT for Illumina RNA UD Indexes (Illumina, San Diego, CA, USA). Samples were processed following the manufacturer’s instructions. The resulting libraries were multiplexed and sequenced with 100 base-pair (bp) paired-end (PE100) reads to a depth of approximately 25 per million reads on an Illumina NovaSeq 6000. Samples were demultiplexed using the bcl2fastq v2.20 Conversion Software (Illumina, San Diego, CA, USA).

### 5.4. Differential Expression Analysis 

The RNA-Seq analysis pipeline consisted of the following steps: quality control using fastp [80] and MultiQC [81], quantification using salmon [82], and differential expression analysis using DESeq2 [83]. R version 4.3.4 and Python version 3.8.5 were used for the data analyses, file management, and visualization. The cutoff value of the adjusted *p*-value (Benjamini–Hochberg method) for multiple testing was predefined as 0.05. The minimum required absolute fold change was set as 1.25 in the log2 scale. All computation was conducted in Amazon Elastic Computing (EC2) instances, the virtual servers in the cloud computing environment. A heatmap was generated after applying hierarchical clustering on the normalized gene expression values with Euclidean distance and complete linkage.

### 5.5. Weighted Gene Correlation Network Analysis (WGCNA) Analysis 

We performed a supervised WGCNA on non-normalized RNAseq data from all subjects. Genes with raw counts of less than 10 in all phenotypic groups (KD, MIS-C, and HC) were excluded. The one-step automatic network construction and module detection was used based on a soft power of 12 and signed topological-overlap matrices. The gene expression profile of a module was summarized by module eigengene, which is defined as the principal component of the module. All genes were univocally assigned to a single module based on their quantified module membership of intramodular connectivity. The analysis was computed using the R package *WGCNA* v1.72 [84].

### 5.6. Statistical Analysis

Values were expressed as medians and interquartile ranges. The Mann–Whitney *U* or Kruskal–Wallis test was used to analyze the differences among the indicated groups. *p* < 0.05 was considered statistically significant. 

## 6. Conclusions

This ex vivo model allowed insight into the disease pathogenesis in an otherwise clinically inaccessible tissue. ECs incubated with pre-treatment sera from patients with KD and MIS-C showed important differences in the transcriptional response. Compared to KD, ECs incubated with MIS-C sera expressed genes favoring cell survival. However, the suppression of genes supporting EC homeostasis and increased serum GC levels in patients with MIS-C may contribute to the transient and quickly reversible myocardial dysfunction that is common in MIS-C patients. ECs incubated with KD sera upregulated genes associated with EndoMT. These transcriptional differences support the clinically observed differences in cardiovascular outcomes in the two diseases.

## Figures and Tables

**Figure 1 ijms-24-12318-f001:**
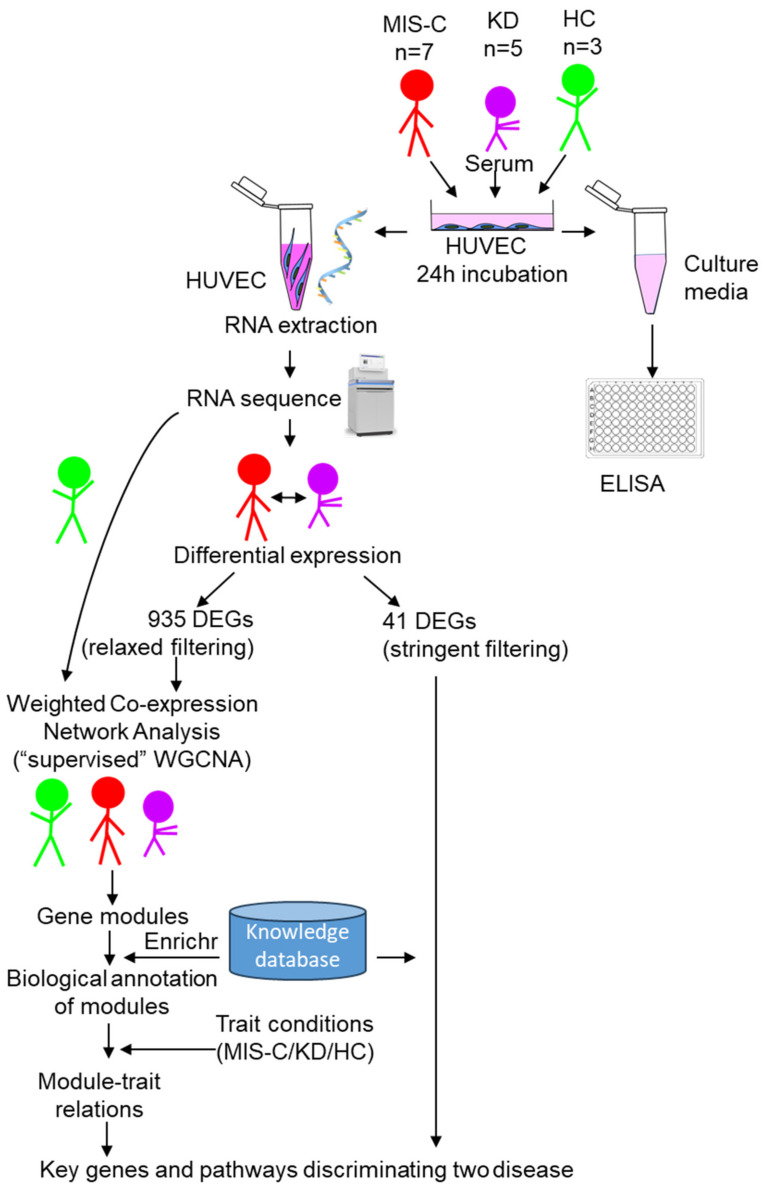
Workflow for EC study. After human umbilical vein endothelial cells (HUVEC) were incubated with sera for 24 h, RNA was extracted from cell lysates, followed by RNAseq, differential expression analysis, and weighted gene co-expression network analysis (WGCNA). Cell culture media from the same experiment were used for ELISA. DEG: differentially expressed genes, Enrichr: gene enrichment analysis (https://maayanlab.cloud/Enrichr/, (accessed on 28 July 2023)).

**Figure 2 ijms-24-12318-f002:**
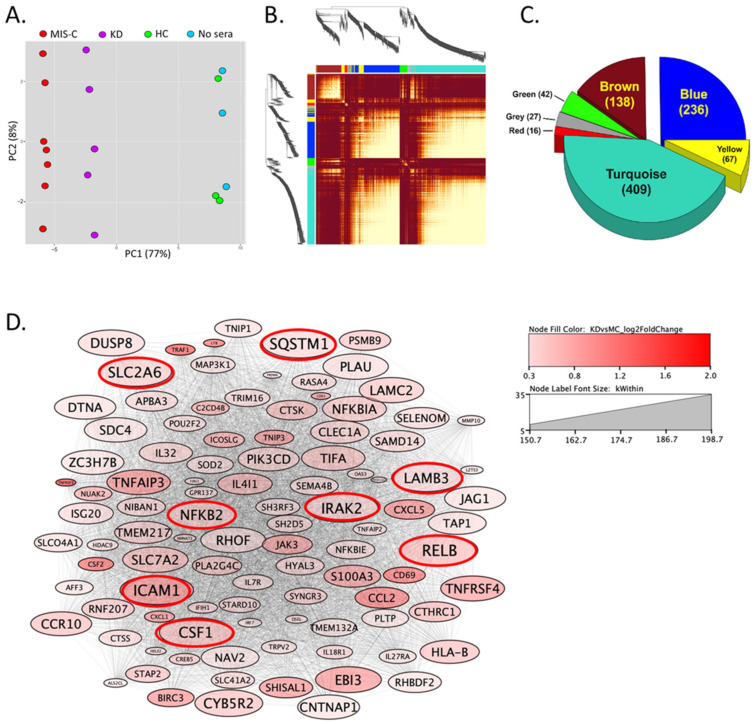
Factor analysis of transcript abundance in cultured ECs incubated with pre-treatment sera from KD, MIS-C, and HC. (**A**) Plot of the first two principal components; (**B**) Pearson correlations between expression profiles of 935 differentially expressed genes (DEGs) between MIS-C and KD with relaxed filter (fold change > 1.1 or <−1.1, unadjusted *p*-value < 0.05, and count ≥ 10). (**C**) Seven modules were identified with weighted gene co-expression network analysis (WGCNA). (**D**) Network analysis was performed using the genes in the turquoise module. Eight of the top ten hub genes (cutoff level: *p*-value of turquoise module membership < 0.05; intramodular connectivity (kWithin) > 150; adjacency > 0.2) are shown in red circles.

**Figure 3 ijms-24-12318-f003:**
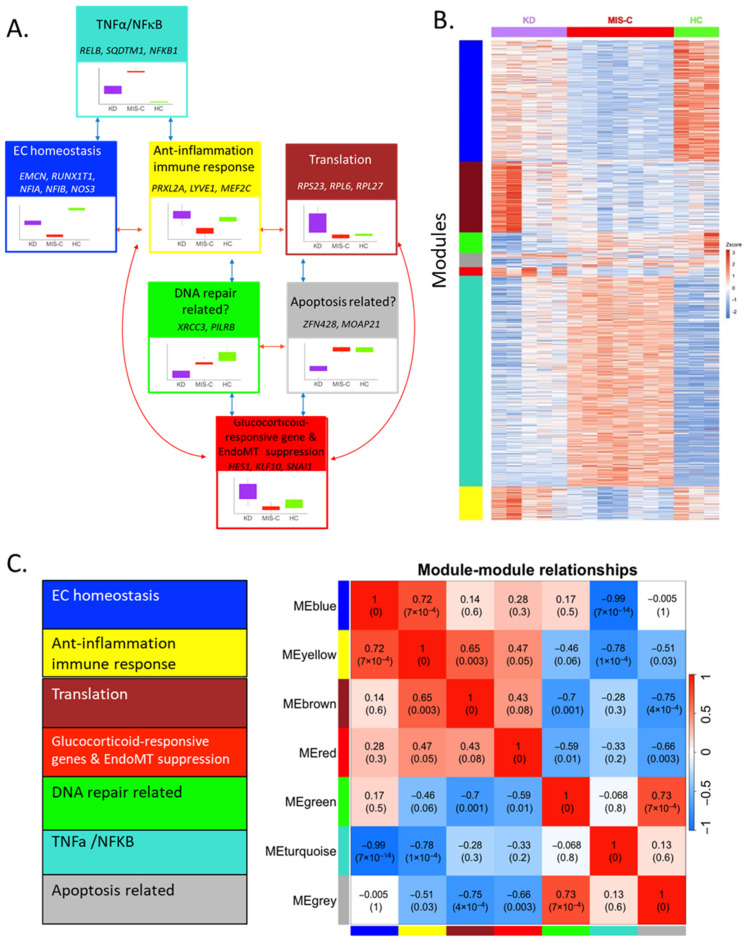
Annotated biological function, correlation, and expression pattern of seven modules in WGCNA: (**A**) Annotation of biological functions shown at the top of each colored box. Genes represent key molecules found by pathway analysis or hub genes. The box plot in each colored box shows the levels of eigengene (Y-axis) from each cultured EC sample incubated with sera from KD, MIS-C, and HC. (**B**) Heatmap of 935 transcripts by seven color-coded modules. (**C**) Module–module correlation plot of eigengenes.

**Figure 4 ijms-24-12318-f004:**
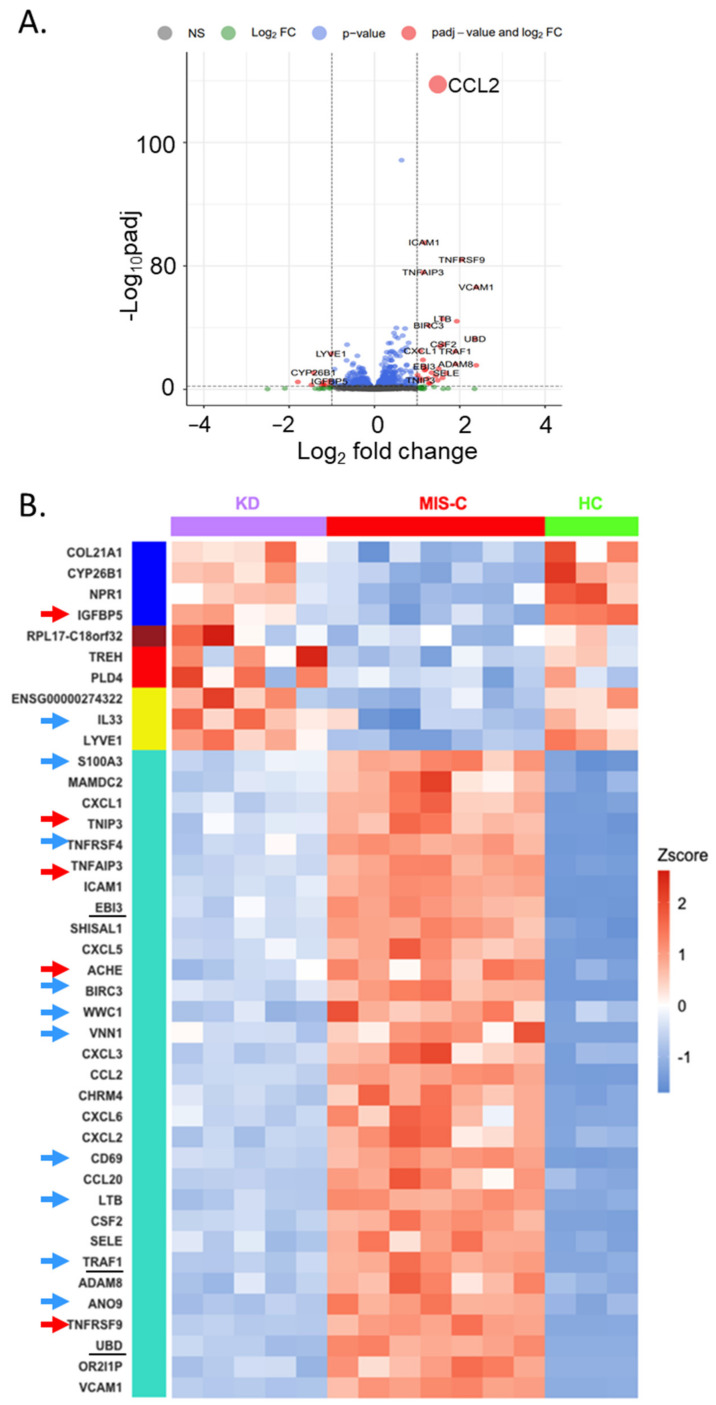
Differentially expressed genes (log_2_ fold difference > 1 and adj *p* < 0.05) between ECs incubated with sera from MIS-C and KD. (**A**) Volcano plot showing differential abundance of transcripts in ECs based on fold difference in log2 scale and adjusted *p*-values. Compared to KD, transcripts to the left of the dotted vertical line were less abundant in ECs incubated with sera from MIS-C patients, while those to the right were more abundant. (**B**) KD vs. MIS-C heatmap with 41 differentially expressed genes (DEGs) using KD (*n* = 5, top purple bar), MIS-C (*n* = 7, top red bar), and HC (*n* = 3, top green bar). The color bars to the left of the heatmap (blue, brown, red, yellow, and turquoise) represent the gene modules classified by WGCNA. The genes with the arrows have pro-apoptotic (red) or anti-apoptotic (blue) effects. The genes with underline bind to SQSTM1.

**Figure 5 ijms-24-12318-f005:**
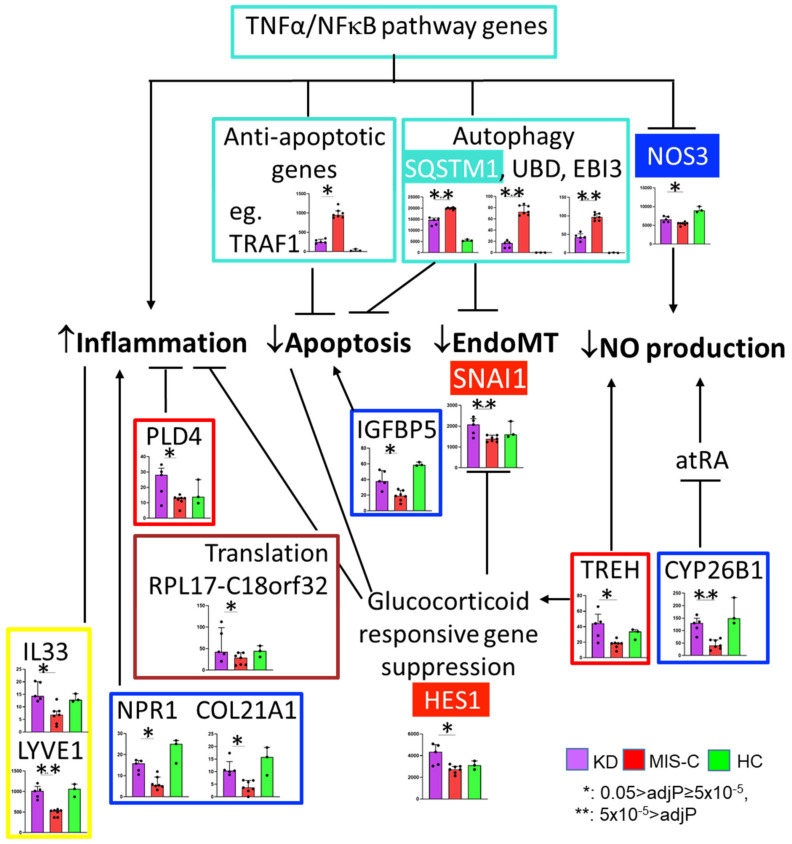
Suggested biological meaning at the intersection of the gene modules from network analysis and 41 DEGs between MIS-C and KD. Color-coded boxes represent the gene modules from WGCNA. Genes in filled boxes were not among the 41 DEGs but were key molecules in the WGCNA modules. SQSMT1, UBD, and EBI3 have important roles in protein degradation [32,33], and SQSTM1-dependent degradation of snail (SNAI1), a transcription factor regulating EndoMT, has been reported [34]. *Cytochrome P450 family 26 subfamily B member 1* (*CYP26B1*) and *trehalase* (*TREH*) are genes that influence NO production in ECs [35,36]. TREH also induces functional confirmation in the glucocorticoid receptor [37]. IGFBP5 relates to apoptosis [38]. Relation with inflammation was reported for PLD4 [39], NPR1 [40], COL21A1 [41], IL33 [42], and LYVE1 [25]. The effects of IL-33 are either pro- or anti-inflammatory, depending on the disease. LYVE1 is important for leukocyte trafficking. Post-transcriptional regulation of CCL2 by GC is well reported [43]. GC binds to GC receptor (GR) in the cytosol and exerts its anti-inflammatory functions by inhibiting expression of cytokines, chemokines, and adhesion molecules in ECs [44]. GC has tissue-specific action on apoptosis [45]. GC inhibits protein synthesis by inhibiting translation initiation and ribosomal synthesis at the levels of transcription, post-transcription, and translation [46,47,48]. *RPL17-C18orf32* is a read-through transcript between *RPL17* (*ribosomal protein L17*) and *C18orf32* (*chromosome 18 open reading frame 32*) genes.

**Figure 6 ijms-24-12318-f006:**
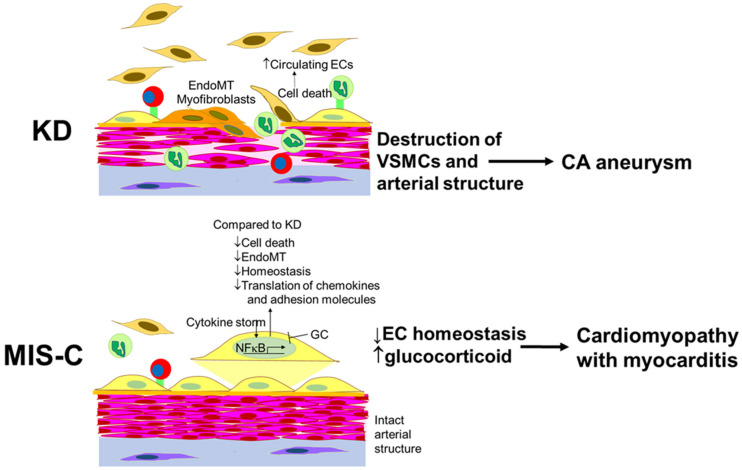
Proposed model of EC response in patients with KD and MIS-C. ECs in MIS-C are more likely to survive and less likely to undergo EndoMT. However, their reduced expression of genes mediating EC homeostasis and increased influence GC may lead to transient LV dysfunction in MIS-C (MIS-C cardiomyopathy/myocarditis) by analogy to Takotsubo syndrome [49]. Although ECs in MIS-C had higher transcript levels of adhesion molecules and chemokines, those protein expressions on the membrane and secretion may be limited, and the endothelium remains intact. Contrastingly, in KD, more ECs may undergo apoptosis and shed into the circulation or undergo EndoMT and promote destruction of the vascular wall [50]. Yellow cells: endothelial cells (ECs); brown cells: circulating ECs; orange cells: myofibroblasts; pink cells: smooth muscle cells; purple cells: fibroblasts; green cells: neutrophils; red cells: monocytes.

**Table 1 ijms-24-12318-t001:** Demographic and clinical characteristics of patients with KD and MIS-C, whose sera were used in the EC experiment.

		MIS-C	KD	HC *^4^	*p* *^5^
		*n* = 7	*n* = 5	*n* = 3	
Age, yrs *^1^		10.5 (8.8–12.2)	1.9 (1.6–3.8)	5.5 (4.2–7.1)	0.005
Male, *n* (%)		6 (86)	3 (60)	2 (67)	NS
Ethnicity, *n* (%)	Asian	0	1 (20)	0	NS
	AA	2 (29)	0	0	
	White	0	1 (20)	2 (67)	
	Hispanic	4 (57)	2 (40)	1 (33)	
	>2 races	1 (14)	1 (20)	0	
Illness day of serum collection *^2^		3 (3–4.5)	5 (4–5)	418 (416–704)	NS
Coronary artery Zmax *^3^		2.1 (1.6–2.7)	3.2 (1.7–3.3)	NA	NS
EF min, %		58 (46–62)	61 (58–66)	NA	NS
Laboratory data	WBC, 10^3^/uL	6.5 (5.0–11.1)	18.4 (12.3–20.7)	NA	0.048
	PLT, 10^3^/mm^3^	140 (93–221)	358 (181–361)	NA	NS
	ESR, mm/h	44 (36.5–53.5)	48 (42–58)	NA	NS
	CRP, mg/dL	21.3 (20.3–26.5)	7.0 (4.8–8.8)	NA	0.048
	Troponin max, ng/mL	0.050 (0.015–0.225)	ND	NA	NA

*^1^: median (interquartile range (IQR)) unless specified. *^2^: Illness Day 1= first day of fever. *^3^: Maximum Z score (internal diameter normalized for body surface area) for the right and left anterior descending coronary arteries. *^4^: Late convalescent sera (illness day 414–990 days) from healthy children with a remote history of KD and with always normal coronary arteries by echocardiography, *^5^: *p*-values calculated by Mann–Whitney test for continuous variables between two groups (MIS-C vs. KD) and Fisher’s exact test for categorical variables. AA: African American, EFmin: the lowest ejection fraction (EF) level during hospitalization, WBC: white blood cell count, PLT: platelet count, ESR: erythrocyte sedimentation rate, CRP: C-reactive protein, Troponin max: the highest troponin level during hospitalization. ND: not done, NA: not applicable, NS: not significant.

## Data Availability

The RNA-Seq data generated during the current study are publicly available in the NCBI GEO with accession number GSE236833.

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
