# Peer review of "Endothelial Cell Response in Kawasaki Disease and Multisystem Inflammatory Syndrome in Children"

_ijms, 2023, doi:10.3390/ijms241512318_

Round 1

Reviewer 1 Report

In this paper Kim and colleagues analyzed the response of cultured ECs to incubation with pre-treatment sera from KD and MIS-C patients, finding several significant differences in terms of gene expression.

The study is well conducted and the results are clearly presented, also with very nice images.

Anyway, in order to help the reader to better understand the scientific hypothesis on which this work is based, I suggest to better clarify which is the aim of the study. In the present way, in fact, the Introduction section does not provide sufficient elements to fully understand what are the Authors searching for.

Author Response

We appreciate a constructive comment to improve this paper. The following sentences were added to Instruction.

“Molecular studies have compared and contrasted KD and MIS-C at the transcriptional and proteomic levels in circulating blood [Ghosh 2022]. Upregulation of inflammatory pathways in both KD and MIS-C have underscored the similarities between the two conditions. We sought to characterize KD and MIS-C at the level of the EC to determine if the EC response might better reflect the divergent clinical outcomes of the two diseases.”

Reviewer 2 Report

Dear authors,

I have Read your interesting work.

I have found it really basilar to know

the different pathogenesis between Kd and Mis in cardiovascolari outcomes

Author Response

We appreciate the reviewer for taking time to review the manuscript and the comment.
